palaeontology, evolution

mammals, Mesozoic, Cenozoic, adaptive radiation, ecomorphological disparity, Theria

**Author for correspondence:**
Gemma Louise Benevento
e-mail: gemma.benevento@gmail.com

# Patterns of mammalian jaw ecomorphological disparity during the Mesozoic/Cenozoic transition

Gemma Louise Benevento[1,2], Roger B. J. Benson[1] and Matt Friedman[1,3]

[1]Department of Earth Sciences, University of Oxford, Oxford OX1 3AN, UK
[2]School of Geography, Earth and Environmental Sciences, University of Birmingham, Edgbaston, Birmingham B15 2TT, UK
[3]Museum of Paleontology and Department of Earth and Environmental Sciences, University of Michigan, 1105 N University Avenue, Ann Arbor, MI 48109, USA

 GLB, 0000-0003-3108-2021; RBJB, 0000-0001-8244-6177; MF, 0000-0002-0114-7384

The radiation of mammals after the Cretaceous/Palaeogene (K/Pg) boundary was a major event in the evolution of terrestrial ecosystems. Multiple studies point to increases in maximum body size and body size disparity, but patterns of disparity for other traits are less clear owing to a focus on different indices and subclades. We conducted an inclusive comparison of jaw functional disparity from the Late Triassic–latest Eocene, using six mechanically relevant mandibular ratios for 256 species representing all major groups. Jaw functional disparity across all mammals was low throughout much of the Mesozoic and remained low across the K/Pg boundary. Nevertheless, the K/Pg boundary was characterized by a pronounced pattern of turnover and replacement, entailing a substantial reduction of non-therian and stem-therian disparity, alongside a marked increase in that of therians. Total mammal disparity exceeded its Mesozoic maximum for the first time during the Eocene, when therian mammals began exploring previously unoccupied regions of function space. This delay in the rise of jaw functional disparity until the Eocene probably reflects the duration of evolutionary recovery after the K/Pg mass extinction event. This contrasts with the more rapid expansion of maximum body size, which occurred in the Paleocene.

## 1. Background

Cenozoic mammals are the 'type' example of an adaptive radiation. Adaptive radiations have assumed a central position in macroevolutionary theory and were originally proposed from a qualitative observation that high taxonomic diversity and ecomorphological disparity of mammals appeared abruptly in the earliest Cenozoic [1]. This has been interpreted as a result of the ecological release of mammals following the extinction of many species, including all non-avian dinosaurs, during the Cretaceous/Palaeogene (K/Pg) mass extinction event [2–4].

Throughout the Mesozoic, representing the first two-thirds of their evolutionary history, mammals were thought to occupy few niches, with taxa of this age often described as small-bodied ecological generalists (e.g. [1,2,5]). Nevertheless, fossil discoveries from the last two decades have revealed greater ecomorphological diversity of inferred locomotor modes than was previously recognized, from as early as the Jurassic. Examples include swimming (*Castorocauda* [6]), digging (e.g. *Docofossor* [7] and *Fruitafossor* [8]), arboreal (e.g. *Agilodocodon* [9] and *Eomaia* [10]) and gliding (e.g. *Vilevolodon* [11], *Maiopatagium* [12] and *Volaticotherium* [13]) mammals, as well as adaptations to carnivory in the badger-sized *Repenomamus* [14]. This high disparity of locomotor modes implies high ecomorphological diversity in general. Indeed, quantitative comparative analyses indicate a major episode of ecomorphological radiation

among early mammals in the Early/Middle Jurassic [15], more than 100 Myr before the extinction of dinosaurs at the end of the Cretaceous. These observations demand a revised appraisal of the ways in which mammals increased their ecomorphological and functional disparity across the K/Pg boundary.

Abrupt and substantial increases in mammalian species richness occurred within the first few million years of the Palaeogene [16–18]. The maximum body size and total range of mammalian body sizes also increased across the K/Pg boundary, initiating in the earliest Paleocene and reaching more modern values in the Eocene [4,16,19,20]. Studies of other aspects of morphological variation, however, have yielded distinct and sometimes contradictory results. Geometric morphometric (GMM) analyses of teeth show decreases in therian disparity during the earliest Palaeogene, on both regional (North America [17,21]) and global [21] scales, whereas dental complexity (orientation patch counts, OPC) suggests that multituberculates underwent a radiation into herbivorous niches at least 20 Myr before the K/Pg boundary [22]. By contrast, discrete character data and reconstructed ancestral states of the entire skeleton (cranial, dental and postcranial) suggest that disparity in eutherian mammals did increase across the K/Pg boundary [23]. Genomic work supports this inference, finding evidence for parallel post-Cretaceous losses of ancestral genes for cuticle-specific digestive enzymes in placentals, which points to the origins of new, divergent trophic roles from insectivorous ancestors in the early Cenozoic [24]. These studies have advanced our understanding of ecomorphological disparity in some groups of mammals and during time intervals adjacent to the K/Pg mass extinction event. However, their focus on individual subclades, varying time scales of investigation and analysis of different types of data limit our ability to detect broader patterns of ecomorphological disparity through time [25] or to compare patterns across the K/Pg boundary to longer 'background' intervals of both the Mesozoic and early Cenozoic. A longer study interval encompassing all mammal groups permits investigation of the impact of events other than the K/Pg on total mammalian ecomorphological disparity. Particularly noteworthy are the Cretaceous Terrestrial Revolution and the Paleocene–Eocene Thermal Maximum, both of which have been implicated as triggering important changes in mammalian evolution (e.g. [26–29]).

In contrast to previous studies, we include species from all clades of Mammaliaformes (used here to refer to all lineages descended from the common ancestor of *Adelobasileus* and living mammals) from the Late Triassic to the end of the Eocene. We focus on a set of six functional ratios made up of nine continuous character traits of the mandible, a proxy for feeding ecology, and one of the axes along which mammals were originally hypothesized to increase their ecomorphological disparity across the K/Pg ([1]; who observed differences in the teeth (feeding ecology) and limbs and feet (locomotor ecology)). The dataset includes 256 species, from which we quantify patterns of jaw functional disparity through time for Mammaliaformes as a whole, as well as for crown Theria. We aim to establish the scope of expansion in mammalian jaw functional morphology and identify how the patterns of jaw morphological disparity compare to patterns of increasing body mass disparity across the K/Pg boundary.

# 2. Material and methods

## (a) Functional data

Six mechanically relevant jaw ratios (relative diastema length, relative molar row length, jaw closing mechanical advantage, jaw slenderness, coronoid process slenderness and relative articulation offset; further described in the electronic supplementary material; and see Anderson *et al.* [30] and Lazagabaster *et al.* [31]) were collected from 256 species representing all major terrestrial mammalian groups from the Late Triassic to the end-Eocene. Jaw measurements were collected from images of jaws in the literature, as well as first hand from fossil mammal collections (listed in the electronic supplementary material). Specimen measurements were recorded using digital callipers correct to 0.1 mm, and images from the literature and specimen photographs were measured using the program GIMP to place landmarks on images. Distances between these landmarks were calculated in R.

Our analysis of mandibular function space occupation begins in the Late Triassic, concurrent with the earliest appearance of Mammaliaformes [5]. Disparity through time, however, was calculated from the Middle Jurassic onwards, because sampling of this interval is sufficient to allow disparity to be calculated reliably. Both analyses continue until the end of the Eocene because ecomorphotypes recognized in the Eocene are suggested to mirror, to a greater or lesser extent, those of modern mammal faunas [2]. Moreover, mammalian maximum body masses reached modern levels by the middle Eocene (approx. 40 Ma [20]). Extending our study until the end-Eocene therefore enabled a more comprehensive investigation of mammalian disparity during the protracted disparity increase following the K/Pg boundary (66 Ma).

## (b) Dating and binning specimens

Time bins used are based on Mesozoic stages and Cenozoic North American Land Mammal Ages (NALMAs), to reflect known changes in mammal faunas through time. Stages and NALMAs were amalgamated where bin length was particularly short, or where sampling of mandibles was low, to enable a more consistent analysis of disparity through time. The shortest time bin included in disparity analyses is 6.5 Myr (Clarkforkian–Wasatchian) and the longest time bin is 24.5 Myr (Aptian–Albian).

One specimen per species was included in our final dataset. Total species age ranges, obtained from Fossilworks (www.fossilworks.org), were assigned to each specimen. Where this age range reflected the genuine species occurrence range, the specimen was included in all appropriate bins. Where age estimates for species known from single specimens or formations span multiple bins owing to uncertainty, ages were drawn randomly from a uniform distribution within the range of uncertainty during each iteration of 5000 bootstrap subsamples for disparity through time estimates (figure 2; electronic supplementary material, figures S6–S8).

## (c) Principal components analysis

All six functional ratios for all 256 mammals were subjected to a principal components analysis (PCA). Where appropriate, proportional ratios were subjected to a logit transformation. This spreads out the tails of the distribution for proportions to correct for a tendency for ratios to not fall within the entire range of theoretically possible values (0–1). All data were then $z$-transformed. This scales all traits to have equal variance and a mean of 0, giving each trait equal weight in the PCA. All analyses were conducted in R. Ordination space was examined by epoch, to determine changes in occupation of function space among our

focal time bins (e.g. [32]). Ordination spaces plotted against finer time bins (matching those used for disparity through time curves (figure 2)) can be viewed in the electronic supplementary material, figures S3–S5.

## (d) Disparity analysis

Disparity, the spread of species in ecomorphological space [32], was calculated for each time bin using all six PC axes. Minimum spanning tree (MST) length was used as the primary measure of disparity [33] and various other indices are presented in the electronic supplementary material, figures S7 and S8. MST length was preferred because it captures the spread of points in space as well as the distances between clusters of points, without being biased by bimodal data. Variance is an unbiased estimator of spread only when data are unimodal, but many of our intervals have bimodal distributions. Therefore, although a commonly used variance-based metric was computed (sum of variance [34,35]), the results are presented only in the electronic supplementary material, figures S7 and S8. Range-based metrics (e.g. sum of ranges) attempt to capture a different element of disparity: how dissimilar the most disparate species are. However, they are highly susceptible to both sample size variation and the occurrence of outliers [36], and therefore may not describe incompletely sampled fossil data in a biologically meaningful way.

The MST was calculated based on the distance between occupied grid cells across function space. PC scores were rounded to the nearest decimal place to obtain these grids (but see the electronic supplementary material, figure S6 for MST length calculated without grid cells). The function *dist2mst*() was used in R [37] to find the MST length. Bins were rarefied so that each bin contained the same number of species in an attempt to mitigate the effects of uneven sampling through time. Data within each bin were rarefied to six species via sampling with replacement for 5000 iterations to produce the disparity curves seen in figure 2. As this number is relatively low, a second disparity curve was produced using fewer time bins but allowing for a higher quota of 15 species (electronic supplementary material, figure S8a). Overall patterns of mammal disparity across the K/Pg boundary were found to be the same at both quotas. The patterns shown in figure 2 (using a quota of six species) are therefore robust. Error bars represent 95% confidence intervals calculated from sampling six species with replacement from each bin 5000 times. Other disparity measures were calculated for comparison and details about these alternative measures and their results can be found in the electronic supplementary material, figures S7 and S8.

## (e) Collection of taxonomic information

Taxonomic information was collected from the primary literature, as well as Fossilworks (www.fossilworks.org). Taxonomic affiliation was based primarily on the phylogenies of Luo ([38]; based on [8,39]), Close *et al.* ([15]; based on [40]) and taxonomic information in Kielan-Jaworowska *et al.* [5] and Rose [41]. The following groups (comprising a mixture of grades and clades) were employed: non-mammalian Mammaliaformes, stem Theria, Australosphenida–Prototheria–Monotremata, Metatheria-Marsupialia, Eutheria-Placentalia, Carnivora and Creodonta, Multituberculata, Pantodonta, Primates and Plesiadapiformes, Rodentia and 'ungulates'. 'Ungulate' is used here to refer to the polyphyletic grouping of the following hoofed mammal clades: Cetartiodactyla, Perissodactyla, Notoungulata, Dinocerata and Mesonychida. These groups were chosen to represent mammals with close phylogenetic affiliation (e.g. Rodentia), and/or ecomorphological attribution (ungulate). For discussion of the placement of taxonomically contested genera or clades, please refer to the electronic supplementary material.

# 3. Results

Mammalian jaw disparity was low in many Mesozoic intervals compared to Cenozoic intervals other than the Paleocene (disparity measured as Euclidean MST length; see Material and methods). Estimated disparity of the Aptian–Albian (Early Cretaceous) is slightly higher than other Mesozic intervals, comparable to that of the early Eocene. Jaw functional disparity for mammals as a whole remained constant across the K/Pg boundary but increased across the Paleocene/ Eocene boundary, and again in the late Eocene (Duchesnean–Chadronian), the final interval of our study (figure 2). Despite no change in disparity for mammals as a whole across the K/Pg boundary, therian mammals underwent consistent interval-to-interval increases from the Campanian to the late Eocene. As early as the Paleocene, therians had entered regions of function space that were left vacant by K/Pg extinctions, as well as those that were never occupied by Mesozoic mammals (figure 1a; high positive values of PC2).

We primarily discuss patterns on the first four principal component axes (PC1–PC4), which together summarize 87% of variance. Further discussion of the traits correlating to each PC axis and exemplars of extreme morphologies associated with positive and negative PC axes scores can be found in the electronic supplementary material.

## (a) Jaw functional disparity of Mesozoic mammals

Jaw functional disparity was consistently low throughout the Jurassic and much of the Cretaceous, with broad overlap between confidence intervals for different Mesozoic time bins (figure 2). The Aptian–Albian is a notable exception, with higher estimated mandibular functional disparity than any other Mesozoic time bin, after rarefaction. Visual assessment of the full sample for each epoch suggests that this overall stability arises from relatively low function space occupation on PCs 1–4 during the Jurassic, followed by slight increases in the Cretaceous (figure 1). Prior to the Late Jurassic, mammals were largely restricted to positive PC1 scores, indicating a predominance of mammals with dorsoventrally slender jaws with longer relative molar rows, little or no diastema, anteroposteriorly wider coronoid processes, and less vertical offset between the jaw hinge and the tooth row for Late Triassic–Middle Jurassic mammaliaforms. *Sinoconodon* from the Late Triassic and the Middle Jurassic haramiyidan *Arboroharamiya* occupied the lower left quadrant of function space alone throughout this time period. From the Late Jurassic onwards, stem therian mammals expanded their occupation into the same areas of negative PC1 function space. This reflects the appearance of multituberculate taxa such as *Liaobaatar*, *Catopsbaatar* (labelled: electronic supplementary material, figure S1) and *Sinobaatar*, which also have dorsoventrally deep jaws with relatively larger diastemas and anteroposteriorly wide coronoid processes.

## (b) Jaw functional disparity of Cenozoic mammals

Early Paleocene mammals occupy a wider range of jaw function space than those of the latest Cretaceous (figure 1a,b). This results from increases in the number of species sampled, and it may reflect a real increase in disparity because mammalian species richness did increase in this interval [17,18]. When rarefied to equal counts of species per time bin, overall mammalian functional disparity is relatively constant across

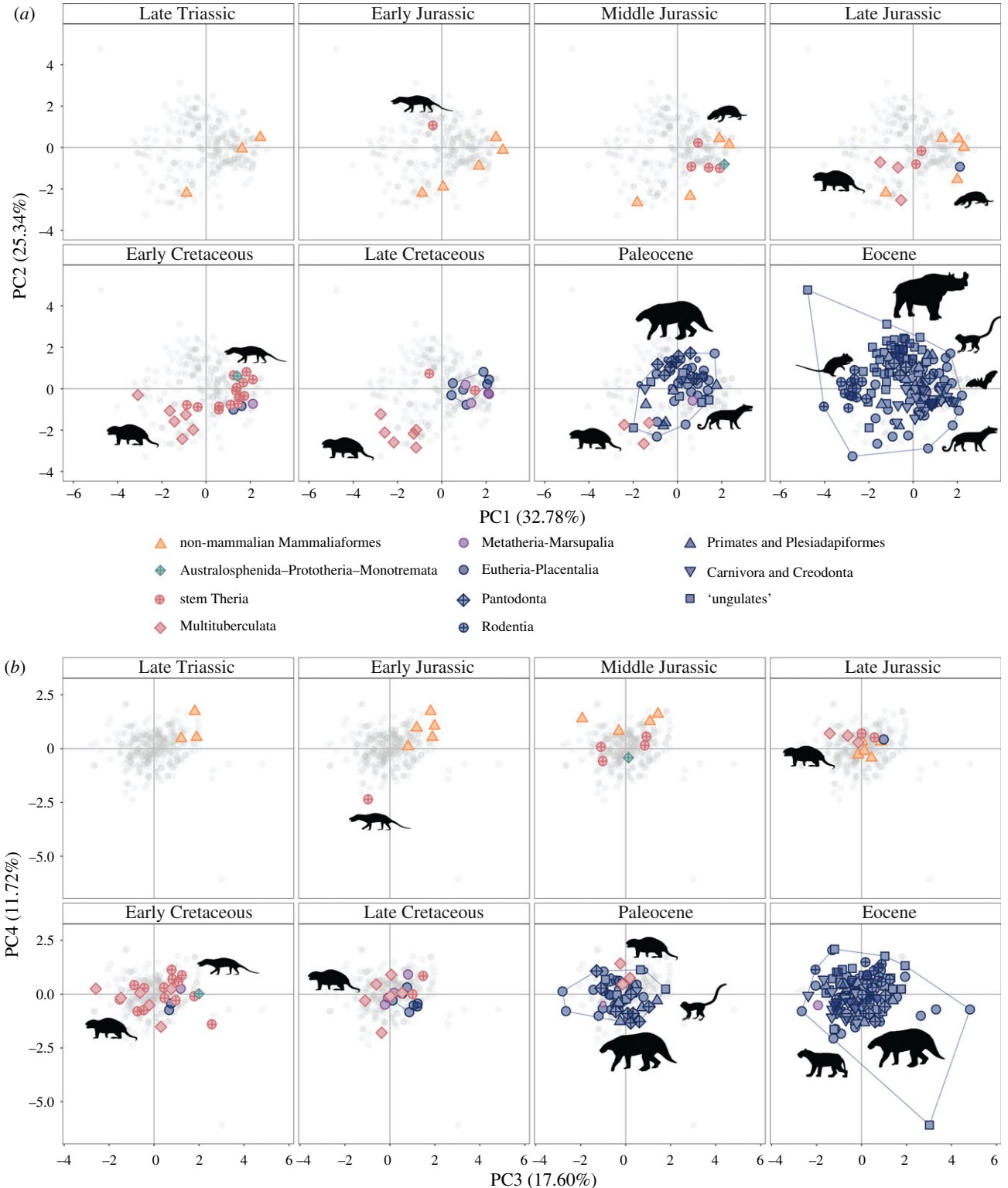

**Figure 1.** Mammaliaform jaw function space from the Late Triassic–Eocene for (a) PC1–PC2 and (b) PC3–PC4. Symbol colour and shape represent taxonomic or ecological groupings (see key within figure). Specimens of uncertain age are plotted as full-size symbols in the most likely bin, and at smaller sizes in other possible—but less likely—intervals. The blue polygon depicts the total spread of eutherian and placental mammals in the Late Cretaceous, Paleocene and Eocene. The electronic supplementary material, figure S1 shows a copy of this figure but with the addition of species labels for mammals that occupy extreme regions of function space across all epochs. The grey points represent the spread of all data points across the total time interval examined. PC5–6 is shown in the electronic supplementary material, figure S2. (Online version in colour.)

the K/Pg boundary (figure 2; purple curve). Disparity increased across the Paleocene/Eocene boundary and then continued to rise throughout the Eocene. Rarefied disparity estimates are higher than any from the Mesozoic by the middle Eocene (figure 2).

Late Cretaceous mammals show a bimodal distribution in function space, with multituberculates occupying a separate region of PC1–2 function space to other mammals from this epoch. This is lost in the Paleocene owing to the appearance of plesiadapiforms such as *Plesiadapis* (labelled: electronic supplementary material, figure S1) possessing intermediate PC1 morphologies (intermediate jaw depths and diastema lengths). The novel appearance of jaws that plot more positively on PC2 during the Paleocene results

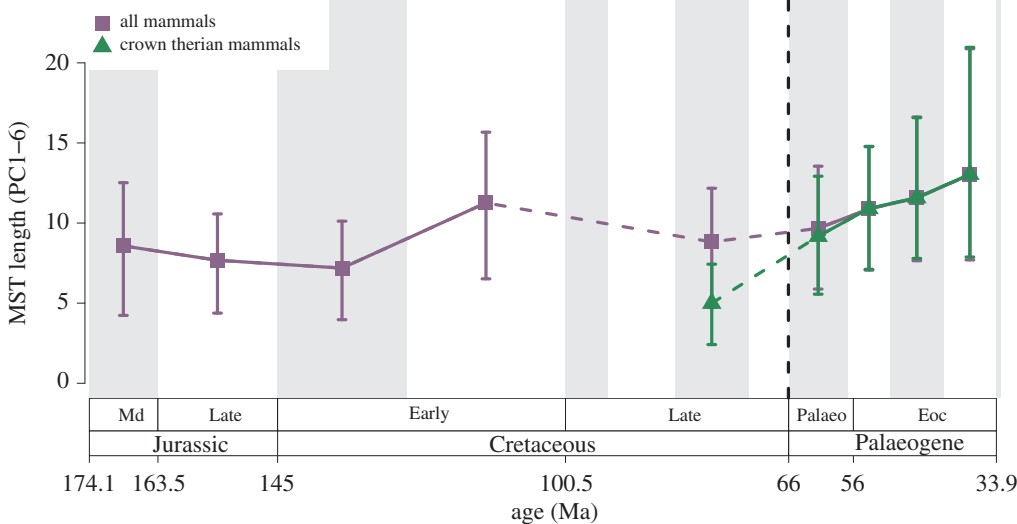

**Figure 2.** Mammaliaform jaw functional MST length through time. The purple curve (square points) represents all mammals, and the green curve (triangular points) represents therian mammals only. Amalgamated stages (Mesozoic) and NALMAs (Cenozoic) with six or more individuals are used as time bins (see Material and methods). Data are rarefied ($n = 6$) and sampled with replacement to enable fairer comparison between bins with differing sample sizes. The 95% confidence intervals represent uncertainty estimated by resampling (procedure described in text). A solid line indicates changes between consecutive time bins. Dashed lines indicate changes across two or more time bin boundaries. Although these curves represent rarefaction to six, total mammal MST length rarefied to fifteen (across fewer bins) can be seen in the electronic supplementary material figure S8a, and analysis using this higher rarefaction number recovers the same pattern across the K/Pg boundary and into the Cenozoic. Curves representing other methods of calculating disparity (see Material and methods) can also be found in the electronic supplementary material figures S7 and S8. (Online version in colour.)

from the appearance of pantodonts, eutherians that represent the first relatively large-bodied herbivorous mammals. Pantodont mandibles are characterized by smaller values of jaw closing mechanical advantage, more anteroposteriorly slender coronoid processes, greater vertical offset between the jaw hinge and tooth row, and larger relative molar row lengths. The Paleocene shows a small reduction in the occupied range of PC1 compared to the Cretaceous. This occurs owing to the disappearance of taxa showing the highest positive PC1 jaw morphologies of the Mesozoic, such as *Sinodelphys*, *Amblotherium* and *Alphadon* (figure 1a; labelled: electronic supplementary material, figure S1). On PC3–4, Paleocene function space occupation is similar to that of the Cretaceous (figure 1b).

The Eocene shows further expansion into areas with more extreme positive values on PC2, general expansion into wider ranges of PC1–2 scores (figure 1a), and population of areas further from the PC3–4 centroids (figure 1b). Among large-bodied herbivorous mammals, perissodactyls and artiodactyls occupy a greater volume of PC2 function space than did Paleocene–Eocene pantodonts. The late Eocene artiodactyl *Hypisodus* shows a particularly divergent ecomorphology relative to other Eocene and earlier mammals. It plots in the uppermost part of the left quadrant in PC1–2 function space (figure 1a) and the lowermost region of the right quadrant in PC3–4 function space (figure 1b), exhibiting a rare combination of traits for pre-Oligocene mammals, including a large diastema, large articulation offset and a slender coronoid process.

## (c) Jaw functional disparity of eutherians

Eutherians constitute an increasingly large subset of total mammalian diversity through the studied interval and show a distinct, expanding pattern of disparity through time from the Late Jurassic to the Eocene (blue in

figure 1a,b: PCs 1–4, and electronic supplementary material, figure S2: PCs 5–6). During this time, they originated in the lower right quadrant and subsequently expanded to lower and more negative PC1 scores and across a wider range of PC2–4 scores, representing the introduction of species with larger diastemas and a dorsoventrally deeper mandibular ramus, as well as species with a larger range of jaw traits associated with PC3–4 (e.g. jaw slenderness, jaw closing mechanical advantage, molar row length and anteroposterior slenderness of the coronoid).

Eutherian mammals appear in our dataset in the Late Jurassic (*Juramaia* [42]), and increase in taxonomic number and proportion between time bins in our dataset until the Eocene. Mesozoic eutherians are almost exclusively confined to regions of function space already occupied by non-mammalian Mammaliaformes and stem Theria on PCs 1–4, but in general do not overlap with the stem therian group Multituberculata. The Paleocene is the first time eutherians explore function spaces not previously occupied by any other mammal groups, as well as areas previously populated exclusively by multituberculates. On PC1–2, pantodonts (including *Leptolambda* and *Alcidedorbignya* (labelled: electronic supplementary material, figure S1)) lie in this completely novel area of function space, and plesiadapiforms, the palaeanodont *Ernanodon* and the dinoceratan *Probathyopsis* occupy regions that had previously been occupied almost exclusively by multituberculates throughout the Mesozoic.

During the Eocene, eutherians are responsible for the further spread into areas of function space not occupied by Mesozoic mammals. 'Ungulates' occupy a wide range of PC1–4 scores in the Eocene. Rodents also spread into function space that was sparsely populated before the Cenozoic. Although rodents are missing from our sample, fragmentary remains indicate their presence in the latest Paleocene [43,44], suggesting that this area of function space was probably populated earlier than is shown in figure 1a.

Crown therian disparity increased across the K/Pg boundary (figure 2; green curve). Although the therian crown includes eutherians and metatherians, very few metatherians can be included in our dataset ($n = 8$). Therefore, this increase in crown therian disparity is evidence for an increase in eutherian jaw ecomorphological disparity across the K/Pg boundary.

## 4. Discussion

We find evidence that mammaliaform mandibular functional disparity was consistently low throughout much of the Mesozoic and the Paleocene, increasing across the Paleocene/Eocene boundary and continually through the Eocene to exceed earlier levels for the first time. When examining disparity patterns for therians alone however, disparity increases from the Cretaceous onwards, first across the K/Pg boundary, and then into the Eocene, and across each successive Eocene time bin. A pattern of comparatively low jaw functional disparity during many Mesozoic intervals emerges despite strong evidence that mammals occupied more ecomorphological roles, as evidenced by their diverse locomotor modes and substrate preferences in the Mesozoic [6,9,13,14,38,45,46], and underwent an Early/Middle Jurassic adaptive radiation entailing both dietary and locomotor innovations [15,38].

### (a) Comparing jaw functional disparity patterns to those reported for body size

In contrast to our results showing that overall mammaliaform jaw functional disparity did not increase across the K/Pg boundary, it is well documented that mean and maximum body size did increase significantly from the earliest Paleocene [19,20]. Body size is an important factor affecting jaw function (e.g. absolute jaw closing force, gape width) and digestive physiology. For this reason, it is likely that changes in body size (and commensurate changes in jaw size) resulted in changes in dietary breadth among mammals from early in the Palaeogene, despite the absence of an associated expansion in the range of jaw functional indices used here until the Eocene. Nevertheless, our results indicate that body size disparity increased at a faster rate than jaw functional disparity among post-K/Pg mammals. Insular dwarfism and gigantism, which are also geologically rapid macroevolutionary phenomena, can be accompanied by little or no change in overall morphology [47,48]. Furthermore, size varies to a much greater extent than some other morphological traits among individuals from the same genus [49]. All these observations suggest that organisms may be more readily able to undergo body size evolution than dietary evolution related to tooth shape [21] or jaw functional disparity. Lower relative evolvability of these morphological traits compared to that of body size may explain the at least 10 million year delay in the accumulation of jaw functional disparity among early Cenozoic mammals documented here.

Substantial changes in body size have been detected during the PETM on short timespans (100s of thousands of years [29,50]). Our study lacks the temporal resolution to investigate the short-term effects of the PETM on mandibular disparity but also does not provide evidence of any long-term decreases in jaw functional disparity as a result of this warming event.

### (b) Shifts in Mesozoic multituberculate functional jaw morphology

Despite overall comparatively low mandibular functional disparity among Mesozoic mammals, a noteworthy shift in multituberculate function space occupation during the Cretaceous (figure 1) is recovered. Previous works have also recognized changes to multituberculate ecomorphology throughout the Mesozoic and have reported increasingly herbivorous diets or a shift toward omnivorous diets with a larger reliance on plant-based foods for this clade, beginning at approximately 100 Ma and increasing throughout the Late Cretaceous [22,51]. This shift represents the origin and radiation of cimolodontan multituberculates. From the Late Jurassic to the Late Cretaceous, multituberculates included in our dataset move toward more extreme negative PC1 and PC2 scores representing the appearance of species with larger diastemas and a greater articulation offset (figure 1a). These traits are often associated with more herbivorous diets [31] and are well-developed in many modern browsers and grazers. The consistent pattern recovered by studies measuring different aspects of the feeding apparatus (dental complexity using OPC [22], jaw morphology using GMM analyses [51] and jaw functional morphology using continuous character trait ratios [current study]) lends weight to the hypothesis that multituberculates responded to environmental changes such as the Cretaceous angiosperm radiation [51,52] by adapting multiple aspects of jaw and tooth morphology to a more plant-rich diet.

Despite this broad agreement, previous studies recovered fluctuations in multituberculate [22,51] and overall mammal [51] disparity during the Cretaceous that are not apparent in our results. Those studies recovered a decline in multituberculate or overall mammalian ecomorphological disparity during the late Early and early Late Cretaceous. Disparity then appears to have recovered [51] or increased [22] in the latest Cretaceous. We recover an increase in Mesozoic mammal disparity during the late Early Cretaceous and a decrease from the late Early Cretaceous to the late Late Cretaceous. We also do not see an obvious shift in mandibular functional disparity in non-multituberculate mammals. These discrepancies may result from low sampling of complete jaws available to our study during this time period, or from disagreement between these morphological datasets and ours. Multituberculate teeth are much more numerous than complete multituberculate jaws during the Cretaceous, suggesting that finer scale patterns of multituberculate OPC disparity for the Cretaceous may be more robust. Future discoveries of Cretaceous mammals will provide a test of the early Late Cretaceous decline of mammalian ecomorphology by increasing all sample sizes.

### (c) Crown therian Cretaceous/Palaeogene disparity increase

Although mandibular functional disparity of mammals as a whole did not increase across the K/Pg boundary, therian mammals do show an initial increase during the Paleocene, and further increases in the Eocene. This rise in therian disparity, coupled with an apparent stasis in overall mammalian jaw disparity until the Eocene suggests that therian mammals may have taken advantage of niches once occupied by non-therian or stem-therian mammal lineages that became

extinct at the end of the Cretaceous. We tentatively argue therefore that therians were the primary beneficiaries of the K/Pg extinction event, showing evidence of preferential diversification during the recovery phase. However, this conclusion should be viewed with the caveat that the fossil record of early Cenozoic mammals is geographically (and therefore taxonomically) biased. The records of South American, Australian and Antarctic Cenozoic fossil mammals are poor in comparison to the rich North American record. This bias obscures our understanding of mammalian evolution across these regions, and in doing so may lead to underestimates of mammalian disparity and diversity among groups that were endemic to these regions.

Although crown Theria comprises both eutherian and metatherian mammals, the patterns of therian evolution discussed here are driven almost entirely by eutherians. Despite much lower diversity among modern metatherian (marsupial) mammals in comparison to eutherians (placentals), they are still ecomorphologically diverse, are important today in Australian ecosystems, and were more abundant in South American ecosystems for much of their evolutionary history [53,54]. Metatherian jaws are rare and often fragmentary in the Mesozoic and early Cenozoic fossil record [55]. These fragmentary remains, however, show that metatherians had substantially higher diversity through time than shown here [21,55,56]. It is likely, therefore, that our analysis of mandibular disparity underestimates the role that metatherians have played in increasing mammalian disparity in certain geographical regions, most notably South America and Australia. In light of this, we suggest that disparity analyses drawn from teeth may be more reliable for metatherian mammals (e.g. [21,55]).

Eutherian mammals increased their function space occupation in every consecutive time bin analysed, starting from their first appearance in the Late Jurassic, meaning that Mesozoic eutherian mammals achieved their highest function space occupation in the Late Cretaceous (figure 1a,b). For this reason, it is possible that eutherian mammal mandibular disparity may have continued to increase into the Cenozoic even if this interval had not been punctuated by a mass extinction. However, maximum body size and body size disparity show a clear expansion at 66 Ma [4,19,20] and this has been directly linked to the extinction event [4]. As previously discussed, this expansion in size combined with even a small number of new jaw functional ecomorphologies probably resulted in increased ecological breadth. This suggests that the K/Pg boundary and the early Cenozoic probably does represent a period of unprecedented ecomorphological opportunity for eutherian mammals, consistent with the hypothesis of macroevolutionary ecological release [23,24].

On short geological time scales, therian ecomorphological disparity (calculated using tooth morphology) has previously been reported to decline across the K/Pg boundary [17,21]. The results presented here suggest that these studies capture the initial stages of mammalian ecological recovery after the K/Pg boundary. Biotic recovery after an extinction event does not occur instantaneously, and may even take up to 10 Myr for true recovery and for opportunistic taxa found in the wake of an extinction event to be replaced by taxa exploiting novel or vacated ecologies [57–59]. This lag in ecological recovery probably explains the short-term decrease in the dental disparity of therians, as recovered by Grossnickle and Newham [21] for the first 4.4 Ma of the Cenozoic.

Paleocene therians predominantly explored regions of function space that were previously occupied by stem-therians, indicating a pattern of turnover involving the replacement of non-therians and stem-therians by therians. We suggest therefore that the initial radiation of crown therian mammals resulted not only from ecological release enabled by extinctions of dinosaurs, but also from those of non-therian and stem-therian mammals.

## (d) Total mammaliaform Paleocene/Eocene increase

The total mandibular functional disparity of mammals exceeded Mesozoic levels during the Eocene. This occurred primarily because of continuing increases in therians, which by then had exceeded the total disparity of all Mesozoic mammals. Therians continued to increase in disparity and function space occupation throughout the rest of the Eocene. In particular, Eocene expansion into novel roles was driven by perissodactyls and artiodactyls, which exhibit anteroposteriorly slender mandibular rami, and large articulation offsets.

This study represents, to our knowledge, the first analysis of total mammaliaform ecomorphological disparity throughout the Mesozoic and into the Eocene for traits other than body size. Our analysis of mandibular functional traits demonstrates a more complex picture of turnover and replacement during the mammalian radiation. We further highlight the necessity for studies to analyse different functional traits, as our results show that, while body size diversification took place in the immediate aftermath of the K/Pg extinction event, aspects of mandibular function took substantially longer to diversify.

Data accessibility. The datasets supporting this article have been uploaded as part of the electronic supplementary material.

Authors' contributions. G.L.B. collected and analysed the data, participated in the design of the study and drafted the manuscript; M.F. participated in conceiving and designing of the study and critically revised the manuscript; R.B.J.B. participated in conceiving and designing of the study and critically revised the manuscript. All authors gave final approval for publication and agree to be held accountable for the work performed therein.

Competing interests. We declare we have no competing interests.

Funding. This research was supported by a NERC studentship for G.L.B. on the Oxford DTP in Environmental Research (NE/L0021612/1), and additional support for G.L.B. was provided by the European Research Council (grant agreement 637483, awarded to Richard Butler). Parts of this work were also funded by the European Union's Horizon 2020 research and innovation programme 2014–2018 grant agreement 677774, awarded to R.B.J.B. (European Research Council Starting Grant, TEMPO).

Acknowledgements. We thank Roger Close, Neil Brocklehurst and John Clarke for helpful discussion regarding methods and the provision of some elements of code, and Thomas Halliday for helpful comments on our manuscript. We thank Hilary Ketchum (OUMNH), Judy Galkin (AMNH), Ken Rose (Johns Hopkins University), Amy Henrici (Carnegie Museum), Chris Norris and Dan Brinkman (Yale Peabody Museum), Amanda Millhouse (Smithsonian Institution NMNH), Bill Simpson, Adrienne Stroup and Susumu Tomiya (Field Museum), Thomas Lehmann and Elvira Brahm (Naturmuseum Senckenberg), and Norbert Micklich (Hessisches Landesmuseum Darmstadt) for access to specimens in their care and/or support during collections visits carried out by G.L.B. We are grateful to Zhe-Xi Luo and an additional anonymous reviewer for insightful comments on our manuscript.

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
