## [Reviewer comments · Proceedings of the Royal Society B: Biological Sciences]

Review History

RSPB-2019-0347.R0 (Original submission)

Review form: Reviewer 1 (Zhe-Xi Luo)

Recommendation

Accept with minor revision (please list in comments)

Scientific importance: Is the manuscript an original and important contribution to its field?

Excellent

General interest: Is the paper of sufficient general interest?

Excellent

Quality of the paper: Is the overall quality of the paper suitable?

Excellent

Is the length of the paper justified?

Yes

Should the paper be seen by a specialist statistical reviewer?

No

Do you have any concerns about statistical analyses in this paper? If so, please specify them explicitly in your report.

No

It is a condition of publication that authors make their supporting data, code and materials available - either as supplementary material or hosted in an external repository. Please rate, if applicable, the supporting data on the following criteria.

Is it accessible?

Yes

Is it clear?

Yes

Is it adequate?

Yes

Do you have any ethical concerns with this paper?

No

Comments to the Author

See the attachment of my written review for both editor and for the authors (See Appendix A).

Review form: Reviewer 2

Recommendation

Accept with minor revision (please list in comments)

Scientific importance: Is the manuscript an original and important contribution to its field?

Excellent

General interest: Is the paper of sufficient general interest?

Good

Quality of the paper: Is the overall quality of the paper suitable?

Excellent

Is the length of the paper justified?

Yes

Should the paper be seen by a specialist statistical reviewer?

Yes

Do you have any concerns about statistical analyses in this paper? If so, please specify them explicitly in your report.

No

It is a condition of publication that authors make their supporting data, code and materials available - either as supplementary material or hosted in an external repository. Please rate, if applicable, the supporting data on the following criteria.

Is it accessible?

Yes

Is it clear?

Yes

Is it adequate?

Yes

Do you have any ethical concerns with this paper?

Yes

Comments to the Author

The authors present a straightforward study on jaw mechanical disparity in mammals across the K-Pg extinction boundary. Utilizing a functionally relevant measure of morphological disparity, they show that there was a delay in the expansion of mechanical disparity in mammal jaws after the event, with an increase in disparity not occurring until the Eocene. At the same time, they are able to identify an expansion in the therian mammals specifically, which take over mechanical space from other groups across the boundary, giving greater insights into the structure of the mammalian response to this event.

Overall, I think this is a strong study which increases our understanding of how this group responded to a major mass extinction event. There are just a few places where I had some further questions.

Pg. 6, Ln. 139-143: I think I understand how the binning of specimens/species was done, but want to make sure I am clear. Specimens (not species) were binned based on where they were found. However, if the age of the specimen's location was not known, they then used the midpoint of the species range for that specimen. I am uncertain why they did not use species ranges for everything? Doing it the way they did, species all end up being in single bins unless they specifically have two jaws from different bins. However, if a species is known from multiple bins, but they only have one jaw, then they will be underestimating the range for that species. I would expect this to lead to underestimated disparity levels. Or am I missing something?

Pgs. 9-10, Ln. 221-226: Here the author's state that jaw functional disparity stayed low from the Triassic to Cretaceous, noting that this arose from a restricted range of function-space occupation. However, in the next sentence they state that function space occupation increased during this time. These statements appear contradictory.

Pg. 10, Ln. 238-241: I understand the reasoning behind performing the rarefaction analysis to standardize the number of specimens examined in each time bin. However, I am concerned about the conclusion that the seeming increase in disparity across the boundary when all jaws are examined is false due to it disappearing when the subset is used. I am unsure what mammalian diversity looks like across the boundary, but if mammals actually undergo an increase in diversity after the boundary that may lead to greater jaw disparity itself. Put another way, if jaw disparity increases just because there are a larger number of species contributing to it, than isn't that a biologically relevant signal as well? This isn't an issue with the method necessarily, but just something the authors may want to address in the discussion.

Pg. 10, Ln 244: The authors should make note that the bimodal distribution in the Cretaceous referred to here is detailed in the previous section. I was initially confused about what bimodal distribution was being referred to until I reread the previous section again.

Decision letter (RSPB-2019-0347.R0)

25-Feb-2019

Dear Dr Benevento:

Your manuscript has now been peer reviewed and the reviews have been assessed by an Associate Editor. The reviewers' comments (not including confidential comments to the Editor) and the comments from the Associate Editor are included at the end of this email for your reference. As you will see, the reviewers and the Editors have raised some concerns with your manuscript and we would like to invite you to revise your manuscript to address them.

Research ethics:

Use of animals and field studies:

If your study uses animals please include details in the methods section of any approval and licences given to carry out the study and include full details of how animal welfare standards were ensured. Field studies should be conducted in accordance with local legislation; please

include details of the appropriate permission and licences that you obtained to carry out the field work.

Insufficient sharing of data can delay or even cause rejection of a paper. This is extremely important.

Please submit a copy of your revised paper within three weeks. If we do not hear from you within this time your manuscript will be rejected. If you are unable to meet this deadline please let us know as soon as possible, as we may be able to grant a short extension.

Sincerely,

Professor John Hutchinson, Editor

Proceedings B

Associate Editor

Comments to Author:

Thank you for the opportunity to review this manuscript. This study examines the expansion of mammalian diversity after the end-Cretaceous mass extinction, focusing on feeding morphology. Results show that diversification in feeding characteristics across mammals post-dates both the extinction, and expansion of body size ranges, by many million years; however, eutherian mammals show a morphospace expansion right after the extinction, which might be a factor in their eventual ascendancy.

Both referees strongly complimented the manuscript, endorsing the topic as one of high significance and likely to attract wide interest. However, both also noted several points that would benefit from additional clarification. In addition, Referee 1 noted several specific considerations for furthering or refining the analyses presented, through the inclusion of data from additional taxa representing groups with currently lower sampling.

Considering these recommendations, I would encourage you to submit a revised version of the manuscript. In this revision, please carefully consider the comments provided by the referees, and explain how your revisions have addressed the concerns that were raised.

Thank you once again for your submission. I hope you find the referee comments to provide constructive guidance for revising your report on your study.

Reviewer(s)' Comments to Author:

Referee: 1

Comments to the Author(s)

See the attachment of my written review for both editor and for the authors.

Referee: 2

Comments to the Author(s)

The authors present a straightforward study on jaw mechanical disparity in mammals across the K-Pg extinction boundary. Utilizing a functionally relevant measure of morphological disparity, they show that there was a delay in the expansion of mechanical disparity in mammal jaws after the event, with an increase in disparity not occurring until the Eocene. At the same time, they are able to identify an expansion in the therian mammals specifically, which take over mechanical space from other groups across the boundary, giving greater insights into the structure of the mammalian response to this event.

Overall, I think this is a strong study which increases our understanding of how this group responded to a major mass extinction event. There are just a few places where I had some further questions.

Pg. 6, Ln. 139-143: I think I understand how the binning of specimens/species was done, but want to make sure I am clear. Specimens (not species) were binned based on where they were found. However, if the age of the specimen's location was not known, they then used the midpoint of the species range for that specimen. I am uncertain why they did not use species ranges for everything? Doing it the way they did, species all end up being in single bins unless

they specifically have two jaws from different bins. However, if a species is known from multiple bins, but they only have one jaw, then they will be underestimating the range for that species. I would expect this to lead to underestimated disparity levels. Or am I missing something?

Pgs. 9-10, Ln. 221-226: Here the author's state that jaw functional disparity stayed low from the Triassic to Cretaceous, noting that this arose from a restricted range of function-space occupation. However, in the next sentence they state that function space occupation increased during this time. These statements appear contradictory.

Pg. 10, Ln. 238-241: I understand the reasoning behind performing the rarefaction analysis to standardize the number of specimens examined in each time bin. However, I am concerned about the conclusion that the seeming increase in disparity across the boundary when all jaws are examined is false due to it disappearing when the subset is used. I am unsure what mammalian diversity looks like across the boundary, but if mammals actually undergo an increase in diversity after the boundary that may lead to greater jaw disparity itself. Put another way, if jaw disparity increases just because there are a larger number of species contributing to it, than isn't that a biologically relevant signal as well? This isn't an issue with the method necessarily, but just something the authors may want to address in the discussion.

Pg. 10, Ln 244: The authors should make note that the bimodal distribution in the Cretaceous referred to here is detailed in the previous section. I was initially confused about what bimodal distribution was being referred to until I reread the previous section again.

Author's Response to Decision Letter for (RSPB-2019-0347.R0)

See Appendix B.

RSPB-2019-0347.R1 (Revision)

Review form: Reviewer 1 (Zhe-Xi Luo)

Recommendation

Accept as is

Scientific importance: Is the manuscript an original and important contribution to its field?

Excellent

General interest: Is the paper of sufficient general interest?

Excellent

Quality of the paper: Is the overall quality of the paper suitable?

Excellent

Is the length of the paper justified?

Yes

Should the paper be seen by a specialist statistical reviewer?

No

Do you have any concerns about statistical analyses in this paper? If so, please specify them explicitly in your report.

No

It is a condition of publication that authors make their supporting data, code and materials available - either as supplementary material or hosted in an external repository. Please rate, if applicable, the supporting data on the following criteria.

Is it accessible?

Yes

Is it clear?

Yes

Is it adequate?

Yes

Do you have any ethical concerns with this paper?

No

Comments to the Author

Dear colleagues, Thank you for making the effort to revise the paper in response to review comments and queries. I believe that this is a very useful contribution to the understanding of disparity pattern in the evolution of mammals. Luo

----- Minor corrections on Typos -----

Nonetheless, in reading the revised manuscript, I spotted several typos. These must be corrected. I am not sure that I caught all these - the authors please do another check:

Volaticotheriam -> Volaticotherium

"multituberculates occupying a separate region of PC1-2 function space to other mammals" - I think this would be better changed to "multituberculates occupying a separate region of PC1-2 function space FROM other mammals"

Xhang -> Zhang - in the reference list

Review form: Reviewer 2

Recommendation

Accept as is

Scientific importance: Is the manuscript an original and important contribution to its field?

Excellent

General interest: Is the paper of sufficient general interest?

Excellent

Quality of the paper: Is the overall quality of the paper suitable?

Excellent

Is the length of the paper justified?

Yes

Should the paper be seen by a specialist statistical reviewer?

No

Do you have any concerns about statistical analyses in this paper? If so, please specify them explicitly in your report.

No

It is a condition of publication that authors make their supporting data, code and materials available - either as supplementary material or hosted in an external repository. Please rate, if applicable, the supporting data on the following criteria.

Is it accessible?

Yes

Is it clear?

Yes

Is it adequate?

Yes

Do you have any ethical concerns with this paper?

No

Comments to the Author

The authors have addressed all of my concerns adequately.

Decision letter (RSPB-2019-0347.R1)

01-Apr-2019

Dear Dr Benevento

I am pleased to inform you that your manuscript RSPB-2019-0347.R1 entitled "Patterns of mammalian jaw ecomorphological disparity during the Mesozoic/Cenozoic transition" has been accepted for publication in Proceedings B.

The referee(s) have recommended publication, but also suggest some minor revisions to your manuscript. Therefore, I invite you to respond to the referee(s)' comments and revise your manuscript. Because the schedule for publication is very tight, it is a condition of publication that you submit the revised version of your manuscript within 7 days. If you do not think you will be able to meet this date please let us know.

To revise your manuscript, log into <https://mc.manuscriptcentral.com/prsb> and enter your Author Centre, where you will find your manuscript title listed under "Manuscripts with

Decisions." Under "Actions," click on "Create a Revision." Your manuscript number has been appended to denote a revision. You will be unable to make your revisions on the originally submitted version of the manuscript. Instead, revise your manuscript and upload a new version through your Author Centre.

NB. From April 1 2013, peer reviewed articles based on research funded wholly or partly by RCUK must include, if applicable, a statement on how the underlying research materials – such

as data, samples or models – can be accessed. This statement should be included in the data accessibility section.

[http://datadryad.org/submit?journalID=RSPB&manu=\(Document not available\)](http://datadryad.org/submit?journalID=RSPB&manu=(Document+not+available)) which will take you to your unique entry in the Dryad repository. If you have already submitted your data to dryad you can make any necessary revisions to your dataset by following the above link. Please see <https://royalsociety.org/journals/ethics-policies/data-sharing-mining/> for more details.

Sincerely,
 Professor John R. Hutchinson
 Proceedings B
<mailto:proceedingsb@royalsociety.org>

Associate Editor:

Board Member: 1

Comments to Author:

Thank you for submitting your revised version of your manuscript. Both referees have indicated that all of the points they raised were addressed appropriately. Referee 1 noted a few remaining typographical points to be corrected before finalization of the manuscript. Thank you once again for your contribution to Proceedings B.

Sincerely – Richard Blob, AE

Reviewer(s)' Comments to Author:

Referee: 2

Comments to the Author(s)

The authors have addressed all of my concerns adequately.

Referee: 1

Comments to the Author(s)

Dear colleagues, Thank you for making the effort to revise the paper in response to review comments and queries. I believe that this is a very useful contribution to the understanding of disparity pattern in the evolution of mammals. Luo

----- Minor corrections on Typos -----

Nonetheless, in reading the revised manuscript, I spotted several typos. These must be corrected. I am not sure that I caught all these - the authors please do another check:

Volaticotheriam -> Volaticotherium

"multituberculates occupying a separate region of PC1-2 function space to other mammals" - I think this would be better changed to "multituberculates occupying a separate region of PC1-2 function space FROM other mammals"

Xhang -> Zhang - in the reference list

Decision letter (RSPB-2019-0347.R2)

08-Apr-2019

Dear Dr Benevento

I am pleased to inform you that your manuscript entitled "Patterns of mammalian jaw ecomorphological disparity during the Mesozoic/Cenozoic transition" has been accepted for publication in Proceedings B.

Open Access

Paper charges

Sincerely,

Proceedings B
mailto: proceedingsb@royalsociety.org

Appendix A

Review of Gemma L. Benevento et al. "Patterns of mammalian jaw ecomorphological disparity during the Mesozoic-Cenozoic transition." Manuscript for Proceedings of The Royal Society B.

By Zhe-Xi Luo/UChicago (zluo@uchicago.edu)

Feb 15, 2019

I enjoyed reading this manuscript. It breaks new ground for understanding the mammal evolution in two critical transitions: K-Pg mass extinction and recovery, and the changes in mammalian biota through PETM (Paleocene-Eocene Thermal Maximum). The paper is insightful and has two strong merits.

First, the paper an elegant demonstration that simple metrics can have great insight. It is a wise choice that the authors adopt the linear morphometrics of whole mandible as a system to characterize disparity. The linear metrics of mandibles are far more convenient for sampling more widely, across major jaw types, and for sampling more taxa within each jaw types. By comparison, geometric morphometrics of tooth types (e.g., tribosphenic molars vs multi-cusped multi-rowed molars vs. triconodont molars) is usually limited to comparison within the same type of teeth that have conserved the same landmarks. In theory, "land-mark free" analyses of whole tooth-row OPC or topographic complexity index can overcome this barrier, but the scan-based data, by their nature, can be very labor intensive to acquire, and also more cumbersome for analyses.

A clever choice of what character system (jaws) with what kind of morphometrics (linear dimensions and ratios) has made this study feasible to extend scope of phylogenies, and geological times in which to assess the macroevolution, as seen in disparity patterns. This has led to some important insights, one of which is the first substantive and significant increase in jaw functional disparity start with PETM.

Second, the authors placed the disparity pattern of whole jaws in a broader context of other disparity metrics on other character systems. The their comparative discussion of this paper is insightful. Alternative disparity patterns may convey different macroevolutionary signals. It makes a great sense to discuss that some character systems (say tribosphenic molars, or jaw disparity here) may be more (or less) evolvable than others (e.g. body size).

Disparity studies of mammal macroevolution have recently come to a crossroads – after the development of multiple proxies to disparity in mammals: disparity within types of teeth (e.g., GM analysis of tribosphenic molars, or OPCs of multituberculates), substrate preferences and locomotor modes as proxy to ecology, and the body size. What I really like about this paper is that it is not a simple-minded claim which one of these would be better than other. Rather, it presents a more biologically realistic perspective, to explore what disparity metric is measuring what aspect of macro-evolution. This important nuance has come through in the discussion of this paper. It is refreshing to me as a referee.

This is a significant and important paper that PRSB should publish as soon as possible. It will certainly stimulate further discussion on a paleobiology question of wide interest.

Out of my responsibility as a referee, I want to alert to the authors for them to consider the revision of the manuscript, all of which are relatively easy to accommodate or to be explained away if not feasible.

1- Additional Taxa in Disparity Analysis

Megaconus should be included – Zhou et al. (2013: fig 2, Nature) published a complete restoration of the whole jaw. Dated to be Middle Jurassic around 166-164 ma, Megaconus can augment the Middle Jurassic bin, which was under-sampled by frank admission of the authors. I searched but did not see this taxon in the SM data table, and I think this should be added – if not already.

Didelphodon should be included – Wilson et al. (2016: fig. 2. Nature Communications; also earlier different specimens published by Fox). In the current version of Benevento dataset, metatherians are under-sampled. Further, Didelphodon has an exceptionally thick and robust jaw (and hypertrophied teeth), and is considered to be a durophagy adaptation. Adding Didelphodon can help to expand the sampling.

Ernanodon is mentioned in text (page 12 lines 288), but it is missing from the SM Data Table. If Ernanodon were included in the analysis, but missing from the SM Data Table, it should be added to the Data Table. If not included in the analysis, the authors can get these measurements from L Radinsky and S-Y Ting (1984: fig 1; Journal of Mammalogy paper, download from JSTOR) or from the photographs in S-Y Ting's original monograph (Ting 1987, Palaeontologia sinica. New Series C. No. 24).

My intuitive sense is that the key conclusion that the big rise of jaw disparity through Paleocene-Eocene transition, driven by eutherians, will still hold up. But the obvious missing entries should be added in, during the revision, as a main merit of this paper is its comprehensive sampling. I think all of the PCA plots and disparity curve through geological time can be easily re-generated after adding these data points.

2 - Comparative Strength of Different Metrics of Disparity

I think that the discussion by Benevento et al on relative strength of different metrics of disparity offers a great insight – it is very refreshing. Foremost, the jaw disparity, and molar disparity of tribosphenic therians, and OPC of multituberculates did not rise with the rapid body size increase. Rather, these are de-coupled from the body size increase, as clearly demonstrated by Grossnickle-

Newham paper and the Benevento manuscript here. Characters with immediate relevancy to dietary diversity did not entirely track the body size, A proxy to disparity in a heuristic sense. The large implication is that is a more generalized disparity metric (like body size) and the metrics with specific functional relevance (like jaw shape or talonid basin metrics) can behave differently, in response to external changes in the ecosystem.

The discussion that some disparity traits (like body size) are more evolvable than others (like tribosphenic molars, and jaw shapes) is insightful. This is an excellent point, which was overdue and should have been brought up, much earlier.

In this light, and to further the paper's main point, I suggest some tweaking. Wherever feasible, it would be better to specify disparity by which traits. For example, the more visually appealing ecomorph disparity manifest in divergent locomotor modes, should be specified in the comparative discussion – see my comments on the second paragraph of the introduction/background.

3 – More Intro Context on PETM

There is a big literature on how Paleocene-Eocene Thermal Maximum (PETM) impacted mammal evolution. During PETM, there is a Clarkforkian-Wasatchian body size change, and dramatic changes on continent-scale mammalian biogeography. The authors need to cite some of these patterns (Like K. C. Beard on global mammal dispersals, and the body size change story by folks in U-M Museum of Paleontology, in the intro/background, so the P-E disparity change of mammals have a broader Earth and Climate change context.

Line by Line Comments and Suggestions

Line 39. Abstract. How about change from “This deferment of jaw functional disparity increases until...” To “This **delay in the rise** of jaw functional disparity until...”

Line 62. It would be better that authors can be more specific to mention “...fossil discoveries from have revealed a greater diversity **of locomotor modes that are proxies to ecomorphology**, than previously recognized”.

Lines 63 – 66. The authors should mentioned more examples for each ecomorphological category here. For digging, you should mentioned Docofossor (Luo et al. 2017 Nature), in addition to Fruitafossor. Docofossor (docodont) and Fruitafossor (a crown mammal clade on its own) represent iterative evolution of digging in different clades. A further difference is that Docofossor is hypothesized to be subterranean mammaliaform, while Fruitafossor may not be. For gliding, Vilevolodon (a fossil represented both by jaw and by skin membrane) is a haramiyid glider, and phylogenetically distinctive from Volaticotherium (a eutriconodont).

These ecological specializations evolved (and then re-evolved) in different clades of Mesozoic mammaliaforms.

Line 66-67. How about change to “...indicate a major episode of ecomorphological radiation among early mammals in the Early/Middle Jurassic...”

Line 96. Change to “the end of Eocene”.

Line 119. Regarding “Our study begins in the Middle Jurassic”. This sentence needs to be revised. Although the statement is consistent with Fig 2 (the metric of the minimum spanning tree - starting point at Middle Jurassic) and Figs S6-S7, it is not consistent with Fig 1 PCA plots starting in Late Triassic. This introductory sentence needs to accurately catch the gist of both principal component “ecospace” plots, and the disparity metrics on evolution over geological time.

Maybe you should say “Our analyses of ecospace deployment by mammaliaforms start at the first appearance of Mammaliaformes in Late Triassic (refs). But our statistical analyses on disparity metrics start in the Middle Jurassic, and continue onward through Eocene because sampling is large enough to allow the analyses of disparity metrics from Middle Jurassic and onward.”

Line 125. How about change from “a thorough investigation” to “a more comprehensive investigation”

Line 143. The sentence that “Duplicate species are removed...” is not clear here. Do authors mean to say “duplicate species within the same time bin are removed”?

Line 160 Reference [44] is a wrong reference. It should be [45]?

Line 194. It would be better to explain two terms here, to preempt the frequent complaints from the “police on cladistic correctness” that are out there. How about this - the “stem Mammalia” be rephrased as “stem mammaliaforms (also known as non-mammalian mammaliaforms),” “stem Theria” Would also need to be explained in a similar manner.

Line 201. Change “Despite no change in total mammal disparity” to “Despite no change in disparity for mammals as a whole”

Line 217. How about “exemplars of extreme morphologies”

Line 233. Liaobaatar (Aptian-Albian) and Catopsbaatar (Late Cretaceous) are OK but not the best available examples to argue your point (as you meant here).

The first examples of prominent dorsoventrally deep jaw is the Haramiyidan Megaconus (Zhou et al. 2013) dated to 165-64 ma (Middle Jurassic). Its jaw is

deeper than that of Late Triassic Haramiyavia. Vilevolodon would be a striking example in this kind of morphology for Late Jurassic.

Line 288 – There is an error of omission on Ernanodon. This mammal is mentioned in the main text as a taxon newly occupying multituberculate ecospace. But the taxon is not listed in supplementary material data table that accompanies the PRSB manuscript. Was Ernanodon included in the analysis, but missed out in the data table? If it is included in the analysis, but not in the table, then there is an error.

Lines 297-298. Authors mentioned that the sampling of metatherians was limited. Didelphodon vorax is an important Late Cretaceous metatherian, and it should be included in the analysis. However, it is not listed in Supplementary Materials Data Table. A relative complete skull with full mandible has been reconstructed by Wilson et al. (2016: Nature communications) (see also Richard C. Fox's several papers), and can (and should) be included in the analysis.

Lines 309-312. Locomotor modes and jaw functional morphology are two different aspects of ecomorphological disparity. I suggest the authors to tweak this sentence to make this more clear. Change to “A pattern of comparatively low jaw functional disparity among Mesozoic mammals emerges despite strong evidence that more ecomorphological roles of mammals, **as evidenced by their diverse locomotor modes and substrate preferences in the Mesozoic** [6, 9, 10, 25-31128], and they underwent an Early/Middle Jurassic adaptive diversification by both dietary diversity and locomotor innovations [11, 25].”

Lines 328-329. How about change to “... may be more readily able to undergo body size evolution, than the dietary evolution of tooth shape [17], or jaw functional evolution.”

Lines 372-373. Change to “Although **mandibular functional disparity of mammals as a whole** does not increase across the K/Pg boundary, therians do show an initial increase during the Paleocene”

Lines 394-398. Something is not right with this sentence. Did you mean to say: “It is likely, therefore, that our analyses on jaws underestimate the role that metatherians have played in increasing mammalian disparity in certain geographic regions, most notably South America and Australia. **Possibly the metatherian disparity drawn on teeth would be a more reliable appraisal** [e.g. 17, 37].”

Line 405. Change to “... show a clear expansion **at** 66 Ma...”

Line 419. How about a small tweak here - Change to “... This lag.... likely explains the short-term decrease **in dental disparity of therians, as** recovered by Grossnickle and Newham...”

Lines 550-563 Two additional references should be cited here:

Luo, Z.-X., Q.-J. Meng, Q. Ji, D. Liu, Y.-G. Zhang, and A. I. Neander. 2015. Evolutionary development in basal mammaliaforms as revealed by a docodontan. *Science* 347: 760-764.

Meng, Q.-J., Grossnickle, D. M., Liu, D., Zhang, Y.-G., Neander, A. I., Ji, Q., and Z.-X. Luo. 2017. New gliding mammaliaforms from the Jurassic. *Nature* 548: 291-296. (doi:10.1038/nature23476).

Formatting Inconsistency in following references that should be re-formatted:

Ref. 9 Meng J et al. 2006.

Ref. 14. Benson RBJ et al. 2016.

Ref. 21. Slater GJ and Friscia AR. 2018. Preprint

Ref. 42. Anderson PSL et al. 2013.

Ref. 45. Foot M 1999.

Ref. 46. Gower JC and Ross GJS. 1969.

Ref. 48. Wills MA et al. 1994.

Ref. 51. Luo ZX and Wible JR. 2005.

Line 666. Figure 1 caption. Change from “mammaliforms” to “mammaliaforms”

Line 674. Figure 2 caption. Change from “mammaliforms” to “mammaliaforms”

Figure 2 caption. The Y-Axis label “MST” Length (PC1-6). “MST = the minimum spanning tree” should be explained in Figure caption.

Appendix B

Department of Earth Sciences
South Parks Road, Oxford OX1 3AN

15th March 2019

Dear John Hutchinson, Zhe-Xi Luo, and our second anonymous reviewer,

We would first like to thank you for your thoughtful consideration of our manuscript. We are grateful for your comments and feel strongly that your suggestions have helped to strengthen our study and completed manuscript.

Below, we outline each point for correction or consideration, and detail what we have done to address these concerns. Additionally, our revised manuscript has been resubmitted with all edits highlighted in red. All figures, both from the main text and the supplement, have been reproduced using our updated dataset and binning method.

Reviewer 1:

Additional Taxa in Disparity Analysis: Megaconus, Didelphodon, and Ernanodon

Megaconus, *Didelphodon*, and *Ernanodon* have all now been added to our analyses. We are grateful to Zhe-Xi Luo for noticing these missing taxa. *Megaconus* and *Ernanodon* were already in our dataset, but had been subset from the final analyses due to missing data in our taxonomic grouping column. *Didelphodon* was an oversight and has now been measured and added to our dataset and analyses.

Comparative Strength of Different Metrics of Disparity: Wherever feasible, it would be better to specify disparity by which traits

We have read through our manuscript and added information about the type of disparity measured each time we mention patterns of disparity through time.

More Intro Context on PETM: There is a big literature on how Paleocene-Eocene Thermal Maximum (PETM) impacted mammal evolution. During PETM, there is a Clarkforkian-Wasatchian body size change, and dramatic changes on continent-scale mammalian biogeography. The authors need to cite some of these patterns (Like K. C. Beard on global mammal dispersals, and the body size change story by folks in U-M Museum of Paleontology, in the intro/background, so the P-E disparity change of mammals have a broader Earth and Climate change context

Mention of the PETM and its effects on mammal evolution have been added to the Introduction and Discussion as follows:

Intro: 'A longer study interval permits investigation of the impact of events other than the K/Pg on mammalian ecomorphological disparity. Particularly noteworthy are the Cretaceous Terrestrial Revolution and the Paleocene-Eocene Thermal Maximum, both of which have been implicated as triggering important changes in mammalian evolution [e.g. 26-29].'

Discussion: 'Substantial changes in body size have been detected during the PETM on short timespans (100's of thousands of years [29, 51]). Our study lacks the temporal resolution to investigate the short-term effects of the PETM on mandibular disparity but also does not provide evidence of any long-term decreases in jaw functional disparity as a result of this warming event.'

Line 39. Abstract. How about change from "This deferment of jaw functional disparity increases until..." To "This **delay in the rise** of jaw functional disparity until..."

This line now reads 'This delay in the rise of jaw functional disparity until...' as suggested.

Line 62. It would be better that authors can be more specific to mention "...fossil discoveries from have revealed a greater diversity **of locomotor modes that are proxies to ecomorphology**, than previously recognized"

This line now reads ‘...greater ecomorphological diversity of inferred locomotor modes than was previously recognised’

Lines 63 – 66. The authors should mentioned more examples for each ecomorphological category here.

Docofossor was added as an example of a digging mammal, *Agilodocodon* as an arboreal mammal, and *Vilevolodon* and *Maiopatagium* as gliding mammals.

Line 66-67. How about change to “...indicate a major episode of ecomorphological radiation among early mammals in the Early/Middle Jurassic...”

This line now reads ‘...indicate a major episode of ecomorphological radiation...’ as suggested.

Line 96. Change to “the end of Eocene”

This line now reads ‘...from the Early Jurassic to the end of the Eocene’

Line 119. Regarding “Our study begins in the Middle Jurassic”. This sentence needs to be revised. Although the statement is consistent with Fig 2 (the metric of the minimum spanning tree - starting point at Middle Jurassic) and Figs S6-S7, it is not consistent with Fig 1 PCA plots starting in Late Triassic. This introductory sentence needs to accurately catch the gist of both principal component “ecospace” plots, and the disparity metrics on evolution over geological time.

This line has been edited to read ‘Our analysis of mandibular function space occupation begins in the Late Triassic, concurrent with the earliest appearance of Mammaliaformes [5]. Disparity through time, however, was calculated from the Middle Jurassic onward, because sampling of this interval is sufficient to allow disparity to be calculated reliably. Both analyses continue until the end of the Eocene because ecomorphotypes recognised in the Eocene are suggested to mirror, to a greater or lesser extent, those of modern mammal faunas [2].’

Line 125. How about change from “a thorough investigation” to “a more comprehensive investigation”

This line now reads ‘a more comprehensive investigation’ as suggested.

Line 143. The sentence that “Duplicate species are removed...” is not clear here. Do authors mean to say “duplicate species within the same time bin are removed”?

This section has been re-written to reflect our revised approach to binning our data, in response to comments from reviewer two. Please see below for more information on how our data is now binned.

Line 160 Reference [44] is a wrong reference. It should be [45]?

Edited to the appropriate reference.

Line 194. It would be better to explain two terms here, to preempt the frequent complaints from the “police on cladistic correctness” that are out there. How about this - the “stem Mammalia” be rephrased as “stem mammaliaforms (also known as non-mammalian mammaliaforms),” “stem Theria” Would also needs to be explained in a similar manner.

We have re-phrased ‘stem Mammalia’ to ‘non-mammalian mammaliaforms’ throughout our document and figures, as we agree that this is a more accurate description of the early mammaliaforms included in this study. After careful consideration we feel happy, however, that ‘stem Theria’ is the appropriate term to describe all of those mammals more closely related to crown therian mammals (eutherians + metatherians) than to any other mammal clade.

Line 201. Change “Despite no change in total mammal disparity” to “Despite no change in disparity for mammals as a whole”

This line has been edited to ‘Despite no change in disparity for mammals as a whole across the K/Pg boundary’ as suggested.

Line 217. How about “*exemplars of extreme morphologies*”

This line has been edited to ‘exemplars of extreme morphologies’ as suggested.

Line 233. *Liaobaatar (Aptian-Albian) and Catopsbaatar (Late Cretaceous) are OK but not the best available examples to argue your point (as you meant here). The first examples of prominent dorsoventrally deep jaw is the haramiyidan Megaconus (Zhou et al. 2013) dated to 165-64 ma (Middle Jurassic). Its jaw is deeper than that of Late Triassic Haramiyavia. Vilevolodon would be a striking example in this kind of morphology for Late Jurassic.*

This has been edited to read ‘Prior to the Late Jurassic, mammals were largely restricted to positive PC1 scores, indicating a predominance of mammals with dorsoventrally slender jaws with longer relative molar rows, little or no diastema, anteroposteriorly wider coronoid processes, and less vertical offset between the jaw hinge and tooth row for Late Triassic - Middle Jurassic mammaliaforms. *Sinoconodon* from the Late Triassic, and the Middle Jurassic haramiyidan *Arboroharamiya* occupied the lower left quadrant of function space alone throughout this time period. From the Late Jurassic onwards, stem therian mammals expanded their occupation into the same areas of negative PC1 function space. This reflects the appearance of multituberculate taxa such as *Liaobaatar*, *Catopsbaatar* (labelled; Fig. S1) and *Sinobaatar*, which also have dorsoventrally deep jaws with relatively larger diastemas and anteroposteriorly wide coronoid processes.’

This is edited to give further information on which mammals plot in the lower left quadrant of function space through time. Although *Megaconus* does have a deeper jaw than many of the mammals known from the same time period, it does not plot in the lower left quadrant, the same region of function space that multituberculates occupy. *Sinoconodon* and *Arboroharamiya*, however, do.

Line 288 – There is an error of omission on *Ernanodon*. This mammal is mentioned in the main text as a taxon newly occupying multituberculate ecospace.

Ernanodon has been re-added to our analysis, and the statement that it occupies similar function space to the multituberculates is true.

Lines 297-298. Authors mentioned that the sampling of metatherians was limited. *Didelphodon vorax* is an important Late Cretaceous metatherian, and it should be included in the analysis. A relative complete skull with full mandible has been reconstructed by Wilson et al. (2016: Nature communications) (see also Richard C. Fox’s several papers), and can (and should) be included in the analysis.

Didelphodon vorax has been added to our dataset. This mammal was initially omitted because GLB was unsure what fossil material the lower jaw reconstruction from Wilson et al, 2016 was based upon. We now feel confident, based on Luo’s suggestion, that this is a reliable reconstruction and have measured this specimen.

Lines 309-312. I suggest the authors to tweak this sentence to make this more clear. Change to “A pattern of comparatively low jaw functional disparity among Mesozoic mammals emerges despite strong evidence that more ecomorphological roles of mammals, as evidenced by their diverse locomotor modes and substrate preferences in the Mesozoic’.

This line has been edited to read ‘...as evidenced by their diverse locomotor modes and substrate preferences in the Mesozoic’ as suggested.

Lines 328-329. How about change to “... may be more readily able to undergo body size evolution, than the dietary evolution of tooth shape [17], or jaw functional evolution.”

This line has been edited to read ‘...may be more readily able to undergo body size evolution than dietary evolution related to tooth shape [21] or jaw functional disparity.’

Lines 372-373. Change to “Although *mandibular functional disparity of mammals as a whole* does not increase across the K/Pg boundary, *therians do show an initial increase during the Paleocene*”

This line now reads ‘Although mandibular functional disparity of mammals as a whole did not increase across the K/Pg boundary...’ as suggested.

Lines 394-398. Something is not right with this sentence. Did you mean to say: “It is likely, therefore, that our analyses on jaws underestimate the role that metatherians have played in increasing mammalian disparity in certain geographic regions, most notably South America and Australia. *Possibly the metatherian disparity drawn on teeth would be a more reliable appraisal [e.g. 17, 37].*”

This has been edited to read ‘It is likely, therefore, that our analysis of mandibular disparity underestimates the role that metatherians have played in increasing mammalian disparity in certain geographic regions, most notably South America and Australia. In light of this, we suggest that disparity analyses drawn from teeth may be more reliable for metatherian mammals [e.g. 21, 56].’

Line 405. Change to “... show a clear expansion *at 66 Ma...*”

This line now reads ‘... show a clear expansion at 66 Ma...’ as suggested.

Line 419. How about a small tweak here - Change to “... This lag... likely explains the short-term decrease *in dental disparity of therians, as recovered by Grossnickle and Newham...*”

This line now reads ‘likely explains the short-term decrease in the dental disparity of therians, as recovered by Grossnickle and Newham’ as suggested.

Formatting Inconsistency in references and the addition of two references.

References have been properly formatted and the additional references added to the list.

Line 666/Line 674. Figure 1 and 2 caption. Change from “mammaliforms” to “mammaliaforms”

Edited to ‘mammaliaforms’.

Figure 2 caption. The Y-Axis label “MST” Length (PC1-6). “MST = the minimum spanning tree” should be explained in Figure caption.

Caption now reads ‘Mammaliaform jaw functional minimum spanning tree (MST) length through time.’

Reviewer 2:

Pg. 6, Ln. 139-143: I think I understand how the binning of specimens/species was done, but want to make sure I am clear. Specimens (not species) were binned based on where they were found. However, if the age of the specimen’s location was not known, they then used the midpoint of the species range for that specimen. I am uncertain why they did not use species ranges for everything? Doing it the way they did, species all end up being in single bins unless they specifically have two jaws from different bins. However, if a species is known from multiple bins, but they only have one jaw, then they will be underestimating the range for that species. I would expect this to lead to

underestimated disparity levels.

We are particularly grateful to reviewer two for this suggestion. As a result we have revised the way that we bin specimens, and we feel that this has strengthened our analyses. We now use one specimen per species, and place that specimen into bins based on species ranges. We also randomized bins for every 5000 bootstraps for species that are imprecisely dated. The chance of a species being placed in a given bin is directly related to the amount of overlap between the species range and each bin.

This new approach to binning taxa has resulted in an overhaul in our defined time bins. This was because some time bins consistently had taxa that moved between them (e.g. the Aptian and the Albian) and therefore were amalgamated (now Aptian-Albian).

There has also been a drop in the number of unique specimens used in our analysis (from 271 to 256). This is because subsequent bins sometimes contained different specimens of the same species. This has now been replaced with one well-preserved specimen used across all bins for that species. The result is fewer individual specimens but in general larger overall sample sizes per bin.

Pgs. 9-10, Ln. 221-226: Here the author's state that jaw functional disparity stayed low from the Triassic to Cretaceous, noting that this arose from a restricted range of function-space occupation. However, in the next sentence they state that function space occupation increased during this time. These statements appear contradictory.

These sentences have been revised to read 'Jaw functional disparity was consistently low throughout the Jurassic and much of the Cretaceous, with broad overlap between confidence intervals for different Mesozoic time bins (Fig. 2). The Aptian-Albian is a notable exception, with higher estimated mandibular functional disparity than any other Mesozoic time bin, after rarefaction. Visual assessment of the full sample for each epoch suggests that this overall stability arises from relatively low function space occupation on principal components (PCs) 1-4 during the Jurassic, followed by slight increases in the Cretaceous (Fig. 1).'

Pg. 10, Ln. 238-241: I understand the reasoning behind performing the rarefaction analysis to standardize the number of specimens examined in each time bin. However, I am concerned about the conclusion that the seeming increase in disparity across the boundary when all jaws are examined is false due to it disappearing when the subset is used. I am unsure what mammalian diversity looks like across the boundary, but if mammals actually undergo an increase in diversity after the boundary that may lead to greater jaw disparity itself. Put another way, if jaw disparity increases just because there are a larger number of species contributing to it, than isn't that a biologically relevant signal as well? This isn't an issue with the method necessarily, but just something the authors may want to address in the discussion.

In response to this comment, we have edited our manuscript to acknowledge that a true increase in diversity may well produce meaningful increases in disparity. We have edited the following sentences to read 'Early Paleocene mammals occupy a wider range of jaw function space than those of the latest Cretaceous (Fig. 1a,b). This results from increases in the number of species sampled, and it may reflect a real increase in disparity since mammalian species richness did increase in this interval [17, 18).'

Pg. 10, Ln 244: The authors should make note that the bimodal distribution in the Cretaceous referred to here is detailed in the previous section. I was initially confused about what bimodal distribution was being referred to until I reread the previous section again.

This line has been edited to read 'Late Cretaceous mammals show a bimodal distribution in function space, with multituberculates occupying a separate region of PC1-2 function space to other mammals from this epoch.'

Department of Earth Sciences
South Parks Road, Oxford OX1 3AN

We would once again like to thank our editor and reviewers for their time and effort in commenting on, and providing guidance on our manuscript. We hope that you feel happy that we have addressed your concerns thoroughly and agree that these changes have strengthened our manuscript.

Kind regards,

Gemma Louise Benevento
Department of Earth Sciences, University of Oxford, Oxford OX1 3AN, UK